# Applications of Machine Learning in Palliative Care: A Systematic Review

**DOI:** 10.3390/cancers15051596

**Published:** 2023-03-04

**Authors:** Erwin Vu, Nina Steinmann, Christina Schröder, Robert Förster, Daniel M. Aebersold, Steffen Eychmüller, Nikola Cihoric, Caroline Hertler, Paul Windisch, Daniel R. Zwahlen

**Affiliations:** 1Department of Radiation Oncology, Kantonsspital St. Gallen, 9007 St. Gallen, Switzerland; 2Department of Radiation Oncology, Kantonsspital Winterthur, Brauerstrasse 15, Haus R, 8400 Winterthur, Switzerland; 3Department of Radiation Oncology, Inselspital, Bern University Hospital, University of Bern, 3012 Bern, Switzerland; 4University Center for Palliative Care, Inselspital, Bern University Hospital, University of Bern, 3012 Bern, Switzerland; 5Competence Center for Palliative Care, Department of Radiation Oncology, University Hospital Zurich, 8091 Zurich, Switzerland

**Keywords:** machine learning, artificial intelligence, palliative care, deep learning, natural language processing, response prediction, data annotation, mortality prediction

## Abstract

**Simple Summary:**

To investigate the adoption of machine learning in palliative care research and clinical practice, we systematically searched for published research papers on the topic. We found several publications that used different kinds of machine learning in palliative care for different use cases. However, on average, there needs to be more rigorous testing of the models to ensure that they work well in different settings.

**Abstract:**

**Objective:** To summarize the available literature on using machine learning (ML) for palliative care practice as well as research and to assess the adherence of the published studies to the most important ML best practices. **Methods:** The MEDLINE database was searched for the use of ML in palliative care practice or research, and the records were screened according to PRISMA guidelines. **Results:** In total, 22 publications using machine learning for mortality prediction (n = 15), data annotation (n = 5), predicting morbidity under palliative therapy (n = 1), and predicting response to palliative therapy (n = 1) were included. Publications used a variety of supervised or unsupervised models, but mostly tree-based classifiers and neural networks. Two publications had code uploaded to a public repository, and one publication uploaded the dataset. **Conclusions:** Machine learning in palliative care is mainly used to predict mortality. Similarly to other applications of ML, external test sets and prospective validations are the exception.

## 1. Introduction

While the number of publications leveraging machine learning (ML) techniques has increased in recent years, this increase seems not evenly distributed across different specialties. Advances such as frameworks that made convolutional neural networks (CNNs) easy to train and deploy have mainly given rise to new ways of analyzing images, which have favored fields such as radiology or pathology where images make up a large share of the data [1]. In addition, more ML software applications have been developed in fields and ecosystems where there is more commercial potential for such an application [2].

Palliative care has been largely unaffected by ML developments, even though there are several scenarios where better models could be useful, such as predicting survival or predicting response to and quality of life during palliative therapy [3].

The purpose of this review was therefore to search the literature for publications that use ML techniques explicitly to improve palliative care practice or research and to assess their adherence to the most important ML best practices. The goal was to create a resource that can be used as a starting point for researchers who want to conduct their own ML research in palliative care as well as to highlight interesting developments and issues that should be addressed by future publications.

## 2. Methods

### 2.1. Literature Search

The review was conducted according to the Preferred Reporting Items for Systematic Reviews and Meta-Analyses (PRISMA) guidelines [4].

Original articles that used any kind of machine learning technique to support clinical palliative care practice in humans were included. No constraints regarding language or year of publication were applied. The Medical Literature Analysis and Retrieval System Online (MEDLINE) database was searched on 7 February 2022 via the PubMed interface.

The query was designed to include studies with either the words “palliative” or “palliation” in the title or abstract as well as at least one word indicating the usage of an ML technique in the title. The complete search query that was used was therefore: ((automated[title]) OR (computer aided[title]) OR (computer-aided[title]) OR (CAD[title]) OR (radiomic[title]) OR (radiomics[title]) OR (texture analysis[title]) OR (texture analyses[title]) OR (textural analysis[title]) OR (textural analyses[title]) OR (deep learning[title]) OR (machine learning[title]) OR (ML[title]) OR (neural network[title]) OR (NN[title]) OR (artificial intelligence[title]) OR (AI[title])) AND((palliative[title/abstract]) OR (palliation[title/abstract])) AND (“1950/01/01”[Date-Create]: “2022/02/07”[Date-Create]).

After the exclusion of duplicates, the titles as well as abstracts were screened, and only relevant publications proceeded to full-text screening. The decision as to whether a study met the inclusion criteria of the review was made by two authors (E.V. and P.W.) without the use of automated tools. A third author (N.S.) acted as a referee in case of a potential disagreement between the two authors responsible for screening. All articles that did not focus on the use of ML to support clinical palliative care practice or research were excluded.

The review had not been registered beforehand, and no protocol had been published.

### 2.2. Data Extraction

Two authors (E.V. and P.W.) independently extracted data and discussed any discrepancies. Data were extracted with regard to:Study parameters: Title, authors, year of publication, recruitment period, number of patients in the respective sets, split, and design;Clinical parameters: Task, ground truth, and features that were used for prediction;ML parameters: Target metric, model, software, and hardware;Disclosures: Code availability, data availability, conflict of interest, and sources of funding.

## 3. Results

The inclusion workflow is depicted in Figure 1. The query returned 63 publications and no duplicates. When screening the records, 40 articles were excluded. A complete list of the excluded articles and the respective reasons for exclusion is provided in Appendix A. Eighteen articles were excluded due to not having an ML focus. Ten articles were excluded due to not being original articles. Twelve articles were excluded due to not focusing on palliative care in humans.

Of the remaining 23 articles, all but one by Barash et al. could be retrieved [5]. All 22 articles that underwent full-text screening were included, and the extracted characteristics from all articles are provided in Appendix A, with selected information being presented in Table 1. Most modeling and validation was carried out retrospectively, with a prospective validation study by Manz et al. being the exception, and the studies were published between 2018 and 2022 [6]. Nine publications contained data from patients with various kinds of diseases. The remaining publications analyzed data from patients with hip fractures, liver malignancies, advanced lung cancer, breast cancer, metastatic colorectal cancer as well as Alzheimer’s disease and related dementias. The most frequent use case was mortality prediction (n = 15) followed by data annotation (n = 5).

### 3.1. Disclosures and Declarations

No study declared a conflict of interest with an obvious concrete relation to the publication. Sixteen publications reported sources of funding for the submitted work, all of which appeared to be government agencies or non-profit organizations.

### 3.2. Data and Code Availability

Two publications had code uploaded to a public repository, and one publication uploaded the dataset [6,15]. Five publications mentioned that the data could be obtained from the corresponding author upon request.

### 3.3. Machine Learning

The studies used a variety of different supervised and unsupervised models such as neural networks, (boosted) tree-based classifiers, support vector machines, and hierarchical clustering. The use of R was mentioned in seven publications compared to nine publications referencing python. Outcome metrics were also heterogeneous, with the most frequent one being the area under the receiver operating characteristic curve, which was used in eight publications. Other outcome metrics included accuracy, the area under the precision–recall curve, sensitivity, specificity, precision, recall, and c-statistics.

### 3.4. Use Case: Machine Learning for Mortality Prediction

Avati et al. used a deep neural network to predict if a patient was going to die within 3–12 months from a consultation [7]. A high likelihood of dying within this short-term period was used as a surrogate to determine if a patient should have been referred to palliative care. The model was trained on electronic health record data without any constraints regarding underlying diseases or patient age. In addition to demographic information, International Statistical Classification of Diseases and Related Health Problems (ICD), Current Procedural Terminology (CPT), and RxNorm concept unique identifier (RXCUI) codes were used as features. The retrospective data of 205,571 patients who met the inclusion criteria were split in an 8:1:1 ratio for training, validation, and testing. The model achieved an Area Under Precision–Recall Curve (AUPRC) of 0.69 and a recall of 0.34 at 0.9 precision. Notably, a high number of false positive predictions (i.e., people who were predicted to die within a year but survived longer) were nonetheless diagnosed with a terminal illness and could have benefitted from palliative care involvement.

Cary et al. used logistic regression as well as a multilayer perceptron to predict 30-day and one-year mortality in patients >65 years treated for hip fracture at an inpatient rehabilitation facility [10].

In addition to demographic features, the authors used eight chronic conditions (stroke, diabetes, liver disease, chronic kidney disease, asthma, and heart disease) as well as the Functional Independent Measure (FIM) score as inputs.

No independent test set was used, but the average performance on the 10-fold cross-validation was reported. Both models, a logistic regression and a multilayer perceptron (a feedforward neural network), exhibited very similar performance with an area under the receiver operating characteristic curve between 0.756 and 0.765 for all tasks.

Goldstein et al. used unsupervised clustering to define prognostic subgroups based on molecular profiles for patients receiving palliative selective internal radiation therapy (SIRT, n = 86) or transarterial chemoembolization (TACE, n = 22) for primary or secondary hepatic malignancies [16].

Manz et al. conducted a prospective validation of an ML algorithm that predicted 180-day mortality in an outpatient oncology cohort [6]. Most technical details of the algorithm development were described in a companion publication by Parikh et al. [28]. The model achieved an area under the receiver operating characteristic curve of 0.89, which was in line with the results of the internal validation on retrospective data.

Wang et al. developed and validated a neural network for mortality prediction in patients with dementia [25]. Notably, in addition to standard demographic parameters, the authors used natural language processing (NLP) to extract information from clinical notes and leverage that information for making predictions. On the validation data, the model achieved an area under the receiver operating characteristic curve of 0.978 (95% CI, 0.977–0.978), 0.956 (95% CI, 0.955–0.956), and 0.943 (95% CI, 0.942–0.944) for 6-month, 1-year, and 2-year mortality, respectively.

Zhang et al. developed a General Machine Learning Pipeline (GMLP) to continuously identify individuals with high short-term mortality in a population [27]. In contrast to other publications, the authors did not use data from electronic health records (EHRs), but administrative claims data such as ICD codes, utilization cost, and patterns as well as demographics for their predictions. Out of several algorithms, an AdaBoost achieved the best performance on the test dataset with an area under the receiver operating characteristic curve of 0.73.

Other mortality prediction publications included and described in Table 1 as well as Appendix A are Berg et al. [8], Blanes-Sevla et al. [9], Elfiky et al. [12], Gensheimer et al. [15], Heyman et al. [18], Lin et al. [20], and Yang et al. [26]. Both Elledge et al. [13] and Nieder et al. [23] validated a model for predicting the survival of patients following palliative radiation for bone metastases published by Alcorn et al. [29].

### 3.5. Use Case: Machine Learning to Support Data Annotation in Palliative Care Research

Durieux et al. used a tandem method of machine learning and human coding to identify connectional silence, i.e., a conversational pause that represents a moment of connection between physician and patient in audio recordings of palliative care consultations in an acute care hospital setting [11]. Most technical details were reported in a companion article by Manukyan et al. [22]

The ML algorithm, a random forest classifier, first predicted conversational pauses which were then passed to a human coder (with ten seconds before and five seconds after the pause for context) who evaluated whether the snippet actually contained a pause (587/1000) and if the pause could be classified as connectional silence. In a second experiment, the authors tried to establish the sensitivity of the pause detection algorithm by using it on 100 min of conversation that contained 41 episodes of connectional silence. All 41 of the episodes were identified by the algorithm. In the companion study, the sensitivity for detecting any kind of pause was lower at 0.908.

The authors concluded that coding the whole dataset without ML support would have taken 61% more time than with the tandem approach.

Macieira et al. used a random forest classifier to transform nursing care plans into variables of a palliative care framework which could then be used in further research [21]. The nursing care plans contained nursing diagnoses, interventions, and outcomes (DIOs). DIOs are groups of a diagnosis (e.g., death anxiety), the intervention that the nursing staff used to address the diagnosis (e.g., active listening), and the outcome that the nursing staff was trying to improve (e.g., anxiety level). The authors trained a model to map each DIO to one of eight categories (family, well-being, mental comfort, physical comfort, mental, safety, functional, and physiological). Two human coders classified 1’000 DIOs, two-thirds of which were used for training, with the remaining third being used for validation. The best model achieved an accuracy of 0.89.

Other examples of machine learning for data annotation include Forsyth et al., who used a conditional random field model to extract symptoms from free-text notes of breast cancer patients receiving paclitaxel-containing chemotherapy, as well as Lee et al., who used natural language processing to identify goals of care conversations in electronic health record notes [14,19].

### 3.6. Use Case: Machine Learning for Predicting Morbidity under Palliative Therapy

Guo, Gao et al. used two machine learning models to predict lung infections in patients undergoing palliative chemotherapy for advanced lung cancer from clinical parameters [17]. While the authors report the superior performance of the neural network compared to the simple logistic regression model (area under the receiver operator curve of 0.897 vs. 0.729), it is unclear if independent test sets were created, how the data were split, and who defined the ground truth.

### 3.7. Use Case: Machine Learning for Response Prediction for Palliative Therapy

Van Helden et al. used hierarchical clustering to predict response, progression-free survival, and overall survival in 52 patients receiving first-line and 47 patients receiving third-line palliative systemic therapy for metastatic colorectal cancer [24]. However, the clustering of the 10 extracted radiomics features did not result in additional predictive value compared to the individual units.

## 4. Discussion

Our review found mortality prediction as the most frequent use case of ML in palliative care. While prognostic scores have existed for a long time, many of these scores require face-to-face consultations for their computation [30,31]. As these consultations are naturally time-consuming and do not scale well when it comes to identifying people in need of palliative care in the broader population beyond the hospital setting, it makes sense to search for automated alternatives. Having a tool running continuously on electronic health records or administrative claims data could be used to prompt providers to consider a palliative care referral in patients where the data indicate a potential need before serious complications occur.

When predicting mortality, it is, however, important to keep in mind that it is only a proxy problem for the *need for palliative care*, which is much harder to define. In an ideal world, models that recommend patients for palliative care referral should not only predict mortality and refer patients based on a somewhat arbitrary cutoff but also try to predict the time to clinical deterioration, which is usually the much more relevant event to determine when palliative care is needed, even though an earlier referral has been shown to provide additional benefit [32,33]. Additionally, ML could also try to predict when other consensus-derived referral criteria are fulfilled [34].

In addition to predicting mortality, which is used to determine if palliative care will be involved, the publications by Guo, Gao et al. on predicting lung infection as well as van Helden et al. on predicting response to systemic therapy indicate how ML can be used once palliative care is already involved. Counseling patients about the likelihood of achieving a response and the expected quality of life is difficult, especially for later lines of palliative–oncologic therapy, and could certainly benefit from models that are able to process a larger number of inputs. However, a model like this would have to prove itself in a randomized controlled trial to ensure that it is actually able to improve outcomes (i.e., make patients live either better or longer). While randomized controlled trials would obviously also be desirable for models that suggest patients for referral to palliative care, the danger associated with false predictions of the latter models is smaller. A “too early” referral to palliative care will hardly harm the patient since it does not stop the patient from receiving additional therapy for his disease. If a model, however, incorrectly predicts severe complications or no response for a therapy that would actually have benefitted the patient, it could cause serious harm, which is why a hurdle for this kind of model needs to be higher.

While the previous use cases mainly relate to improving clinical palliative care, another promising use case is applying ML to data annotation in palliative care research. In addition to the studies by Durieux et al. and Macieira et al. who used ML to find conversational pauses and to transform documentation variables, the approach by Wang et al. is especially promising for palliative care [11,21,25]. On palliative care wards, a lot of different providers work with the patients and document their sessions—usually in an unstructured way—in the electronic health record. As manually coding that data involves a lot of effort and might in some cases be almost impossible, aggregating all of it and passing it on to natural language processing could be a serious improvement. An NLP approach has also been employed by other publications that used it to identify palliative patient cohorts for further research. Lindvall et al. used NLP to identify patients who received a gastrostomy with palliative intent, and Brizzi et al. used NLP on magnetic resonance imaging (MRI) reports to identify breast cancer patients with leptomeningeal disease [35,36].

In contrast to ML publications in radiology, employment or funding by the medtech industry, as well as patent applications, are not present, which reduces the risk of bias at the study level [37]. However, data and especially code sharing could be improved. While reluctance to share data is understandable due to privacy concerns and data protection regulations, uploading code to a public repository or adding it as a supplemental file should almost always be possible and comes with numerous benefits. In addition to giving other researchers inspiration for approaching certain problems, it also helps to clarify many ambiguities regarding the methodology. Whenever it is unclear how the data were split or how hyperparameters were set during model training, any technical reader or reviewer can simply look at the code and figure it out themselves.

Another area with room for improvement that could also be addressed by sharing data is the use of external test sets. While ML in palliative care is still in its infancy, not using external test sets is understandable as researchers first have to train and internally validate models before conducting external validations to see if their models generalize well to unseen data from other institutions. However, there are considerable logistics associated with obtaining external test sets eventually. Therefore, sharing anonymized palliative care datasets increases the chance of other researchers finding a dataset with the variables needed for their predictions without having to go through the lengthy process of establishing collaborations first.

While data sharing is desirable, it can be an unpleasant experience if re-coding large parts of the data is required for an already trained model to handle the new data. Recent research shows worrying signals regarding computerized systems’ capability to use and share data.

Even though laboratory and medication data are currently the most standardized clinical data [38,39], a study by Bernstam et al. shows a low level of interoperability on a minimal dataset consisting of six laboratory values and six medications. The mean inter-vendor interoperability on this minimal dataset was only 20% [40].

The broad application of ML/AI technologies requires the standardization of datasets across the clinical enterprise [41]. The solution for the previously mentioned problem is a broad implementation of semantic interoperability principles, defined as a computer’s capability to share and use data unambiguously [42].

The first step in achieving standardized data collection does not have to depend on any technological solution, electronic health records, or systems vendor. Instead, clinicians and researchers in palliative care could join forces and define their basic clinical terminology and models. Several successful strategies have already been developed in other clinical disciplines. Noteworthy are efforts to standardize pathology datasets guided by the International Collaboration on Cancer Reporting (ICCR) and the development of Common Data Elements in radiology [43,44].

Creating larger datasets by sharing data also enables researchers to select more homogenous groups of patients for model development. While it may be tempting to include all patients treated at a palliative care center for modeling in order to have as many observations as possible, palliative care cohorts tend to be very heterogeneous. A variety of underlying diseases as well as people at different ages and time points in their disease trajectory can make modeling difficult. Larger datasets allow for the creation of more homogeneous groups while ensuring a sufficient number of observations.

Possible limitations at the review level include the fact that only articles with “palliative” or “palliation” in their title or abstract were retrieved by the query, which could have led to articles also trying to predict short-term mortality but not mentioning the keywords being excluded. However, it seems appropriate to assume that any research conducted with the intention to improve palliative care practice or research is likely to use one of the terms somewhere in either title or abstract.

The same applies to publications using ML models but not explicitly referring to them in the title. Only one database was queried, but this limitation is mitigated by the fact that the majority of publications in the field of palliative care appear in PubMed-indexed journals.

## 5. Conclusions

In conclusion, machine learning in palliative care is mainly used to predict mortality, but recent publications indicate its potential for other innovative use cases such as data annotation and predicting complications. Similarly to other applications of ML, external test sets and prospective validations are the exception, but some publications have already started addressing this. In the meantime, the added benefit of ML and especially the ability to generalize to data from a variety of different institutions remains difficult to assess.

## Figures and Tables

**Figure 1 cancers-15-01596-f001:**
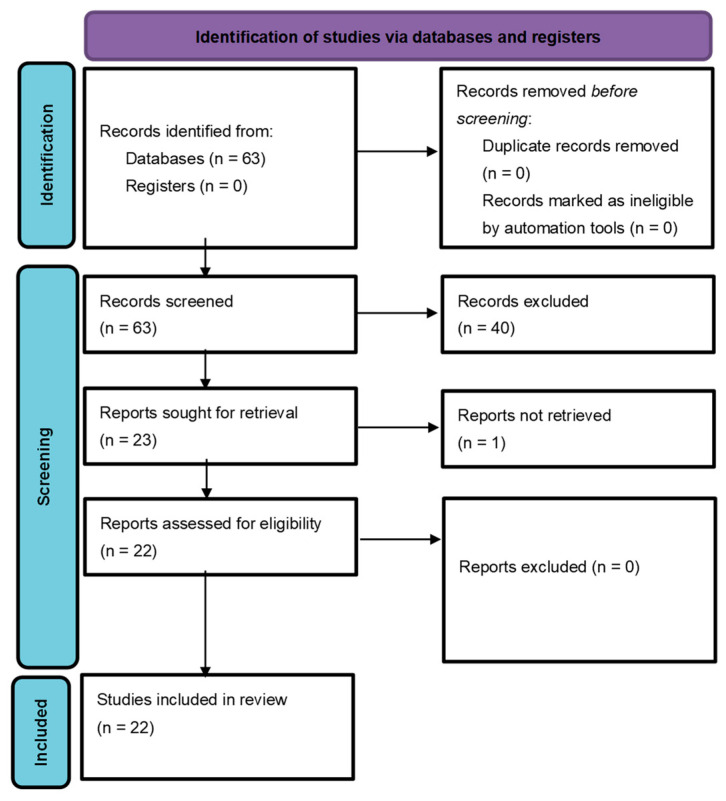
Workflow of the literature search according to PRISMA guidelines. From: Page MJ, McKenzie JE, Bossuyt PM, Boutron I, Hoffmann TC, Mulrow CD et al. The PRISMA 2020 statement: an updated guideline for reporting systematic reviews. BMJ 2021;372:n71. doi: 10.1136/bmj.n71. For more information, visit: http://www.prisma-statement.org/ (accessed on 3 August 2022).

**Table 1 cancers-15-01596-t001:** Summary of extracted study parameters.

Title	Author	Year	Disease	Task
Improving palliative care with deep learning	Avati et al. [7]	2018	Disease agnostic	Predicting mortality within 3–12 months using EHR data from the previous 12 months
Development and validation of 15-month mortality prediction models: a retrospective observational comparison of machine-learning techniques in a national sample of Medicare recipients	Berg et al. [8]	2019	Disease agnostic	Predicting 15-month mortality among community-dwelling Medicare beneficiaries
Design of 1-year mortality forecast at hospital admission: A machine learning approach	Blanes-Selva et al. [9]	2021	Disease agnostic	Predicting 1-year mortality for patients admitted to a hospital
Machine Learning Algorithms to Predict Mortality and Allocate Palliative Care for Older Patients with Hip Fracture	Cary et al. [10]	2021	Hip fracture	Predicting 30-day and 1-year mortality for patients >65 years treated for hip fractures in inpatient rehabilitation facilities
Identifying Connectional Silence in Palliative Care Consultations: A Tandem Machine-Learning and Human Coding Method	Durieux et al. [11]	2018	Disease agnostic	Predicting conversational pauses in palliative care conversations so that human coders could classify the pauses as connectional or not
Development and Application of a Machine Learning Approach to Assess Short-term Mortality Risk Among Patients with Cancer Starting Chemotherapy	Elfiky et al. [12]	2018	Cancer	Predicting 30-day mortality of cancer patients undergoing chemotherapy
External Validation of the Bone Metastases Ensemble Trees for Survival (BMETS) Machine Learning Model to Predict Survival in Patients with Symptomatic Bone Metastases	Elledge et al. [13]	2021	Cancer	Predicting survival in patients receiving palliative radiation for symptomatic bone metastases
Machine Learning Methods to Extract Documentation of Breast Cancer Symptoms from Electronic Health Records	Forsyth et al. [14]	2018	Breast cancer	Extracting patient-reported symptoms from free-text health records of breast cancer patients receiving chemotherapy
Automated Survival Prediction in Metastatic Cancer Patients Using High-Dimensional Electronic Medical Record Data	Gensheimer et al. [15]	2019	Metastatic cancer	Predicting survival from date of first visit after metastatic cancer diagnosis
Optimal multiparametric set-up modelled for best survival outcomes in palliative treatment of liver malignancies: unsupervised machine learning and 3 PM recommendations	Goldstein et al. [16]	2020	Primary and secondary liver malignancies	Clustering patients with liver malignancies according to their survival probability
Prediction of Lung Infection during Palliative Chemotherapy of Lung Cancer Based on Artificial Neural Network	Guo, Gao et al. [17]	2022	Advanced lung cancer	Predicting lung infections in lung cancer patients undergoing palliative chemotherapy
Improving Machine Learning 30-Day Mortality Prediction by Discounting Surprising Deaths	Heyman et al. [18]	2021	Disease agnostic	Predicting 30-day mortality upon emergency department discharge
Identifying Goals of Care Conversations in the Electronic Health Record Using Natural Language Processing and Machine Learning	Lee et al. [19]	2020	Disease agnostic	Identifying goals of care conversation in notes in the electronic health records of patients with a critical illness and/or receiving palliative care
Machine-Learning Monitoring System for Predicting Mortality Among Patients with Noncancer End-Stage Liver Disease: Retrospective Study	Lin et al. [20]	2020	Non-cancer end-stage liver disease	Predicting survival in patients with non-cancer end-stage liver disease
Use of machine learning to transform complex standardized nursing care plan data into meaningful research variables: a palliative care exemplar	Macieira et al. [21]	2021	Disease agnostic	Classifying DIOs (groups of diagnosis, intervention and outcome) into a palliative care framework for hospitalized patients receiving palliative care
Automated Detection of Conversational Pauses from Audio Recordings of Serious Illness Conversations in Natural Hospital Settings	Manukyan et al. [22]	2018	Disease agnostic	Predicting conversational pauses in palliative care conversations so that human coders could classify the pauses as connectional or not
Validation of a Machine Learning Algorithm to Predict 180-Day Mortality for Outpatients with Cancer	Manz et al. [6]	2020	Cancer	Predicting 180-day mortality in an outpatient oncology cohort
Independent Validation of a Comprehensive Machine Learning Approach Predicting Survival After Radiotherapy for Bone Metastases	Nieder et al. [23]	2021	Cancer	Predicting survival in patients receiving palliative radiation for symptomatic bone metastases
Radiomics analysis of pre-treatment [18F]FDG PET/CT for patients with metastatic colorectal cancer undergoing palliative systemic treatment	Van Helden et al. [24]	2018	Metastatic colorectal cancer	Predicting response in patients with metastatic colorectal cancer receiving 1st- or 3rd-line palliative chemotherapy
Development and Validation of a Deep Learning Algorithm for Mortality Prediction in Selecting Patients with Dementia for Earlier Palliative Care Interventions	Wang et al. [25]	2019	Alzheimer’s disease and related dementias	Predicting 6-month, 1-year, and 2-year mortality in patients with Alzheimer’s disease and related dementias
Deep-Learning Approach to Predict Survival Outcomes Using Wearable Actigraphy Device Among End-Stage Cancer Patients	Yang et al. [26]	2021	End-stage cancer	Predicting in-hospital death of end-stage cancer patients on a hospice care unit using wristband actigraphy
Predicting potential palliative care beneficiaries for health plans: A generalized machine learning pipeline	Zhang et al. [27]	2021	12 chronic health conditions	Predicting 1-year mortality in people with certain chronic health conditions from the general population

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
