# Peer review of "Applications of Machine Learning in Palliative Care: A Systematic Review"

_cancers, 2023, doi:10.3390/cancers15051596_

Round 1
Reviewer 1 Report
The following pertinent reports should be mentioned/discussed in the Review:
doi: 10.3390/cancers15020503.
doi: 10.1371/journal.pone.0267012
doi: 10.3390/ijerph19074263.
doi: 10.3389/fpubh.2021.730150.
doi: 10.3389/fendo.2022.1054358.
doi: 10.1371/journal.pone.0253443.
doi: 10.1186/s12967-019-2109-7.
doi: 10.3389/fendo.2022.1056310.
doi: 10.3389/fendo.2022.1030045.
doi: 10.1016/j.jbi.2022.104075
doi: 10.3389/fendo.2022.1083569.
Author Response
N/A
Reviewer 2 Report
The paper is very interesting, as the topic is very hot: how artificial intelligence can help the physician in clinical choice.
It describes the use of machine learning methodology in palliative care.
The authors conducted a systematic review by considering a few keywords and a pub med search.
The authors found 22 relevant articles, which they then commented on.
The introduction outlines the structure of the article sufficiently.
The methods describe the procedure of searching the articles, using prism.
There are some aspects that I think are useful to clarify and motivate before the article is publishable.
I did not understand why the article by Natasha Y, Soffer S et al. Postgrad Med J. 202 Mar;98(1157):166-171.was not retrieved.
Table 1 might contain some information about the sample size of the data from each study.
The conclusions are too short and have as their only finding that the studies are able to predict mortality.
Author Response
Reviewer #2: The paper is very interesting, as the topic is very hot: how artificial intelligence can help the physician in clinical choice.
It describes the use of machine learning methodology in palliative care.
The authors conducted a systematic review by considering a few keywords and a pub med search.
The authors found 22 relevant articles, which they then commented on.
The introduction outlines the structure of the article sufficiently.
The methods describe the procedure of searching the articles, using prism.
Response: We thank the reviewer for the positive assessment of our work.
Reviewer #2: There are some aspects that I think are useful to clarify and motivate before the article is publishable.
I did not understand why the article by Natasha Y, Soffer S et al. Postgrad Med J. 202 Mar;98(1157):166-171.was not retrieved.
Response: The article you mention has a “first published date” (December 3, 2022) and an “online issue publication date” (February 18, 2022). The latter seems to be used by PubMed for indexing and is just after the date of our literature search (February 7, 2022). Therefore, our query was unfortunately unable to retrieve the article.
Reviewer #2: Table 1 might contain some information about the sample size of the data from each study.
Response: We’ve also considered this, but decided to not include it because the information is very heterogeneous and difficult to interpret without context. The range of patients goes from less than 100 to more than 2 million (for the medicare study). Also, not all studies mention the number of patients but instead the number of notes, sentences used for NLP, etc.
Therefore, we prefer to provide that information together with all the other extracted parameters in supplementary table 2. However, if you insist, we can move that information to table 1 as well.
Reviewer #2: The conclusions are too short and have as their only finding that the studies are able to predict mortality.
Response: We have expanded the conclusion:
“In conclusion, machine learning in palliative care is mainly used to predict mortality, but recent publications indicate its potential for other innovative use cases such as data annotation and predicting complications. Similarly to other applications of ML, external test sets and prospective validations are the exception, but some publications have already started addressing this. In the meantime, the added benefit of ML and especially the ability to generalize to data from a variety of different institutions remains difficult to assess.”
Reviewer 3 Report
This was a systematic review of Machine Learning.
Articles were selected according to PRISMA guidelines.
The introduction and goals were succinct and well written.
The search history is well written.
The method of article selection and data extraction is well described.
Different aspects of the study were described – data and code availability, machine learning with a case study of using machine learning for mortality prediction. Another helpful application was the use of machine learning to support data annotation in palliative care research.
The lack of industry interest in this research as a reason for less bias in this research was an interesting observation.
The discussion is good, but sentences could be grouped together to make larger paragraphs. There are many 1 or 2 sentence paragraphs. Perhaps the number of paragraphs could be reduced.
The conclusion is good but perhaps could have its own heading?
Is there a role for machine learning in analysing communication in palliative care e.g. analysis of medical records.
Author Response
Reviewer #2: This was a systematic review of Machine Learning.
Articles were selected according to PRISMA guidelines.
The introduction and goals were succinct and well written.
The search history is well written.
The method of article selection and data extraction is well described.
Different aspects of the study were described – data and code availability, machine learning with a case study of using machine learning for mortality prediction. Another helpful application was the use of machine learning to support data annotation in palliative care research.
The lack of industry interest in this research as a reason for less bias in this research was an interesting observation.
Response:Thank you for the positive assessment of our work.
Reviewer #2: The discussion is good, but sentences could be grouped together to make larger paragraphs. There are many 1 or 2 sentence paragraphs. Perhaps the number of paragraphs could be reduced.
Response: Thank you for this comment. We have reduced the number of 1-2 sentence paragraphs by merging them with other paragraphs.
Reviewer #2: The conclusion is good but perhaps could have its own heading?
Response: We have added a separate heading for the conclusions.
Reviewer #2: Is there a role for machine learning in analysing communication in palliative care e.g. analysis of medical records.
Response: We believe there is. This is indicated by the papers by Manukyan et al. and Durieux et al. that our literature search retrieved. To highlight this, we have extended our conclusion:
“In conclusion, machine learning in palliative care is mainly used to predict mortal-ity, but recent publications indicate its potential for other innovative use cases such as data annotation (even for complex types like recorded conversations) and predicting complications. Similarly to other applications of ML, external test sets and prospective validations are the exception, but some publications have already started addressing this. In the meantime, the added benefit of ML and especially the ability to generalize to data from a variety of different institutions remains difficult to assess.”
Round 2
Reviewer 1 Report
-
Reviewer 2 Report
The authors accurately answered the queries and expanded the conclusions.
In my opinion the paper is complete.